# Rapid assessment of psychological and epidemiological correlates of COVID-19 concern, financial strain, and health-related behavior change in a large online sample

**Benjamin W. Nelson** [1,2,3]*, **Adam Pettitt**[1], **Jessica E. Flannery**[1,2,3], **Nicholas B. Allen**[1]

**1** Department of Psychology, University of Oregon, Eugene, OR, United States of America, **2** School of Medicine, University of Washington, Seattle, WA, United States of America, **3** Department of Psychology, University of North Carolina Chapel Hill, Chapel Hills, NC, United States of America

* bwn@uoregon.edu

**Data Availability Statement:** All study material, code, and deidentified data are available from the

## Abstract

COVID-19 emerged in November 2019 leading to a global pandemic that has not only resulted in widespread medical complications and loss of life, but has also impacted global economies and transformed daily life. The current rapid response study in a convenience online sample quickly recruited 2,065 participants across the United States, Canada, and Europe in late March and early April 2020. Cross-sectional findings indicated elevated anxiety and depressive symptoms compared to historical norms, which were positively associated with COVID-19 concern more strongly than epidemiological data signifying risk (e.g., world and country confirmed cases). Employment loss was positively associated with greater depressive symptoms and COVID-19 concern, and depressive symptoms and COVID-19 concern were significantly associated with more stringent self-quarantine behavior. The rapid collection of data during the early phase of this pandemic is limited by underrepresentation of non-White and middle age and older adults. Nevertheless, these findings have implications for interventions to slow the spread of COVID-19 infection.

## Introduction

In November 2019 the novel severe acute respiratory syndrome coronavirus 2, which causes the coronavirus disease 2019 (COVID-19), emerged in Wuhan, China. Since this time COVID-19 has rapidly spread around the world leading to a pandemic that has so far resulted not only in medical complications and loss of life, but has also led to the largest global economic impact and transformation to daily life, since past global events, such as the Great Depression. As of April 9, 2020 there have been 1,696,139 confirmed cases and 102,669 confirmed deaths [1]. Recently, researchers at the CDC estimated that COVID-19 infectiousness or median R0 is 5.7 (95% CI 3.8–8.9) [2], higher than prior estimates that ranged from 2.0–2.6 ([3]; Note that any R0 value above 1.0 indicates that cases will continue to grow). Furthermore, international COVID-19 case fatality rate (CFR) estimates range from 1.0% to as high as 7.2%

Open Science Framework database (https://osf.io/vtnca/).

**Funding:** This research was supported by funding from Ann Swindells Endowment to the University of Oregon. The funding sources had no role in the study design, data collection and analysis, or submission process.

**Competing interests:** All authors declare no competing interests.

in particular countries, such as Italy which resulted from hospital resources becoming over-whelmed [4]. For comparison, the common flu has an R0 of 1–1.5 and a CFR of .01%. Overall, some initial epidemiological models based on an "unmitigated epidemic" (e.g., absence of individual behavior change and systemic control measures) that doesn't account for the potential of overwhelmed healthcare systems have predicted that despite R0 and CRP estimates, there could be up to 1.1–1.2 million deaths in the United States and up to 250,000 deaths in Great Britain [3].

## Mental health during a global pandemic

While the psychological fallout of past epidemics, such as SARS and Polio have been documented [5], currently, there is a lack of psychological literature directly related to global *pandemics* [6]. The last pandemic, the 1918 influenza pandemic, occurred prior to modern psychological science. Therefore, the potential mental health effects of COVID-19 might be gleaned from other areas of inquiry including 1) primary effects of epidemic disease outbreaks, as well as the secondary effects of 2) economic recessions/depressions, and 3) loneliness, quarantine, and social isolation.

**Disease outbreaks.** Prior epidemics have consistently led to increased mental health difficulties [5]. For example, Polio symptoms and treatment conditions led to trauma [7]. Moreover, the Australian outbreak of equine influenza was associated with psychological distress [8], and the 2003 SARS epidemic in Hong Kong was associated with increased psychological burden, distress, depressive symptoms, fear, and restless sleep [9] as well as elderly suicide [10]. Preliminary research from China on COVID-19 has found higher depression, anxiety, and posttraumatic stress symptoms [11, 12] and non-peer reviewed research has found increased sleep in the United States [13]. Similarly, recent preprints studies have found increased internet mental health-related keyword searches using the Google search engine [14] and an association between social distancing and past-month suicidal ideation and suicide attempts [15].

**Economic recession.** Due to behavioral restrictions on movement around the world to curb the spread of COVID-19, many nations have completely shut down "non-essential" business sectors impacting global economies causing massive disruption [16]. For example, between January 2, 2020 and March 23, 2020 the S&P 500 fell 31.32% and within a single week that ended on March 28, 2020, 6,648,000 Americans filed unemployment claims, which is the highest number of seasonally adjusted initial filed unemployment claims in US history [17], indicating that 10% of the US workforce became unemployed [18]. Some calculations indicate that the United States could see unemployment surpass rates during the Great Depression [19] and the world could see 195,000,000 lost jobs [20] with double digit declines in imports/exports during 2020 likely leading to the largest decrease in world trade since the 2008 Financial Crisis [16].

Economic recessions have reliably been positively associated with mental health degradation and increased negative coping behaviors (e.g., substance use; [21]). For example, experiencing an impact to financial, housing, or employment during the Great Recession of 2008 in the United States was positively associated with increased anxiety, depression, and substance use up to 3–4 years post-recession [21]. Furthermore, country level unemployment, poverty, and foreclosure were positively associated with suicide rates during the 2008 recession [22, 23].

**Quarantine, social isolation, and loneliness.** Quarantine measures of past outbreaks have resulted in higher depression, anxiety, post-traumatic stress symptoms [5], psychological burden, distress, restless sleep [9], and suicide among the elderly, possibly due to increased

disconnection and loneliness [10]. Furthermore, social isolation and loneliness have a negative impact on physical health that is on par with well-known behavioral health variables (e.g., physical inactivity, obesity, substance abuse), such that loneliness, social isolation, and living alone are positively associated with a 26%, 29%, and 32% increased risk for mortality, respectively [24].

Overall, there are objective threats to societal health that COVID-19 poses, which indicate that there is a pressing need to characterize the current impact the COVID-19 pandemic is having on the general public's mental health, financial concern, and to identify variables that are positively associated with successful adherence to self-quarantine recommendations.

### Current study

The current study was cross-sectional and used open materials, including code and deidentified data, available on Open Science Framework (https://osf.io/vtnca/) to 1) characterize whether current levels of individual transdiagnostic mental health symptoms (i.e., anxiety and depression) are elevated when compared to historical normative data, 2) determine whether individual differences in transdiagnostic mental health symptoms or epidemiological data indicating objective risk (e.g., cumulative country and world cases) explain more variance in psychological concern about COVID-19, 3) identify associations between financial difficulties, mental health symptoms, and COVID-19 concern, and 4) delineate whether transdiagnostic mental health symptoms, COVID-19 concern, or epidemiological data explain more variance in the degree of adherence with self-quarantine.

## Methods and materials

### Participants

Due to the time sensitive nature of identifying the initial impact of COVID-19, the current study utilized a rapid response design based on a convenience online sample. The study recruited 2,443 participants around the world to participate in a survey between March 19, 2020 and April 10, 2020. The final sample included 2,065 participants (see Fig 1) after limiting participation to predominantly English speaking countries and geographic regions for which we had a sizable sample (n > 100 per region), which led to selecting participants from the United States (n = 1683), Canada (n = 137), and Europe (n = 245). In addition, participants were removed if they did not meet age criteria of being 18 or older (n = 12), if they did not successfully complete an attention check (n = 130), or if they had an invalid IP address (n = 4). For breakdown of participants by individual countries see S1 File. Inclusion criteria required that participants had to be 18 years old or older (mean age = 34.40 years, SD = 11.49, Range 18–77 years; see S1 File). Our sample identified as predominantly White (80.19%), Non-Hispanic (90.07%), and female (69.20%). The most common household income was $20,000-$49,000 and the most common level of education was some college or higher. See Fig 2 below for full demographic break down of race, ethnicity, gender, political orientation and health. See Fig 3 below for a demographic breakdown of Income, Employment, Financial Strain, and Health Behavior Change. The study was approved by the University of Oregon Institutional Review Board.

### Recruitment

Participants were recruited through promoted social media ads on Twitter (n = 403) and Facebook (n = 36), Instagram (n = 6), and the survey was submitted to a call for Covid-19 related studies on Reddit (n = 970) [25]. Smaller numbers of participants were recruited through

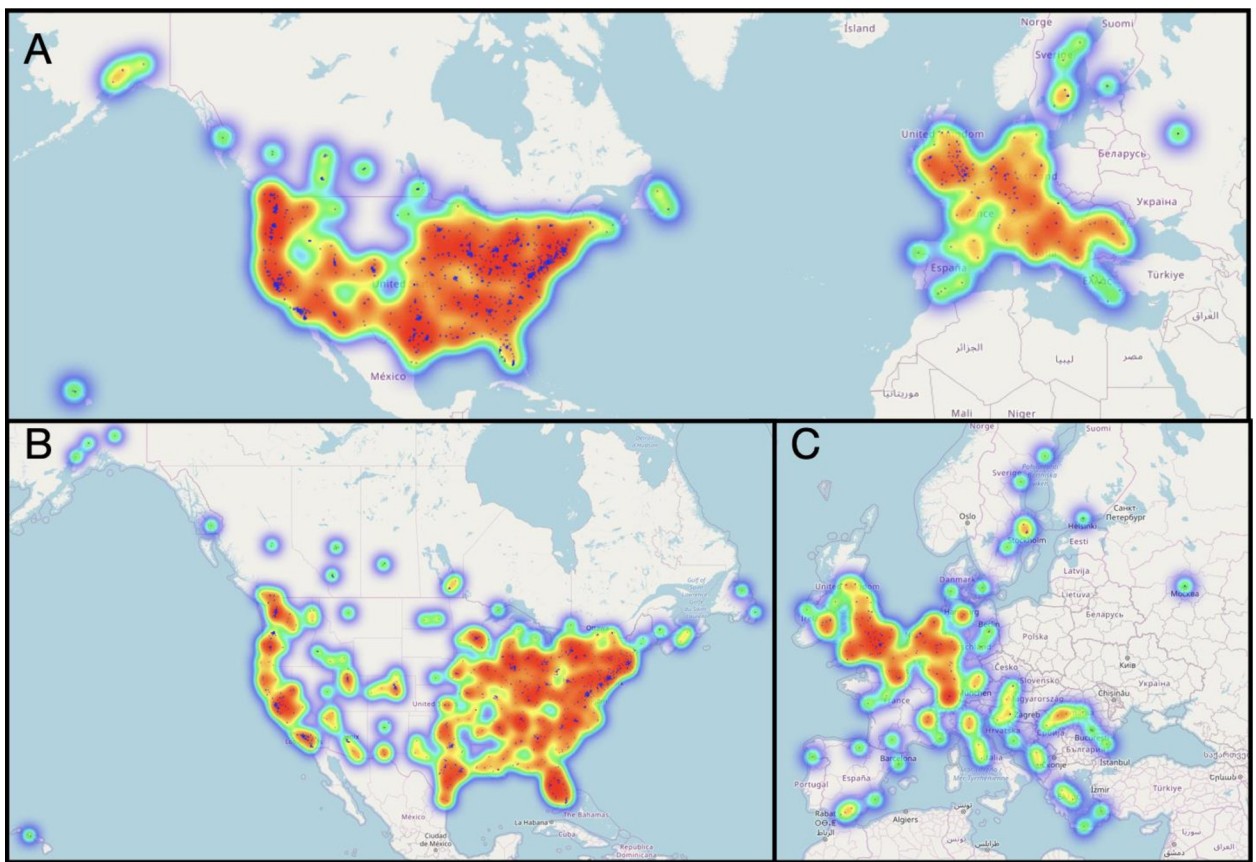

**Fig 1. Participant location.** A. United States, Canada, and Europe, B. Zoomed Image of United States and Canada, C. Zoomed Image of Europe.

word of mouth (n = 16, i.e., by sharing the study link with friends), other methods (n = 150), and 484 participants did not provide a response regarding their method of recruitment.

## Assessment procedures

Participants consented via Qualtrics and were then asked to complete a set of questionnaires (see OSF for materials https://osf.io/vtnca/) to assess current transdiagnostic symptoms of mental health, demographics, COVID-19 related behaviors, and COVID-19 concern. The questionnaire was completed in a median of 5 minutes and 18 seconds. Participants were not compensated for survey completion.

## Measures

**Symptom measures.** *Generalized Anxiety Disorder-2 (GAD-2).* The GAD-2 has been shown to be a valid measure of anxiety symptoms [26–28]. The GAD-2 consists of the first two questions of the GAD-7 has been shown to be just as informative as the GAD-7 for identifying GAD and other anxiety disorder diagnosis [29]. A cut-off score $\geq$ 3 has been established for identifying likely Generalized Anxiety Disorder with a sensitivity (i.e., true positive rate) value of 0.86 (0.76–0.93) and a specificity (i.e., true negative rate) value of 0.83 (0.80–0.85) [27], although a more recent meta-analysis identified a lower pooled sensitivity (i.e., true positive rate) value of 0.80 (0.62–0.91) and a lower pooled specificity (i.e., true negative rate) value of 0.81 (0.65–0.91) [28]. In a sample of over 5,000 individuals

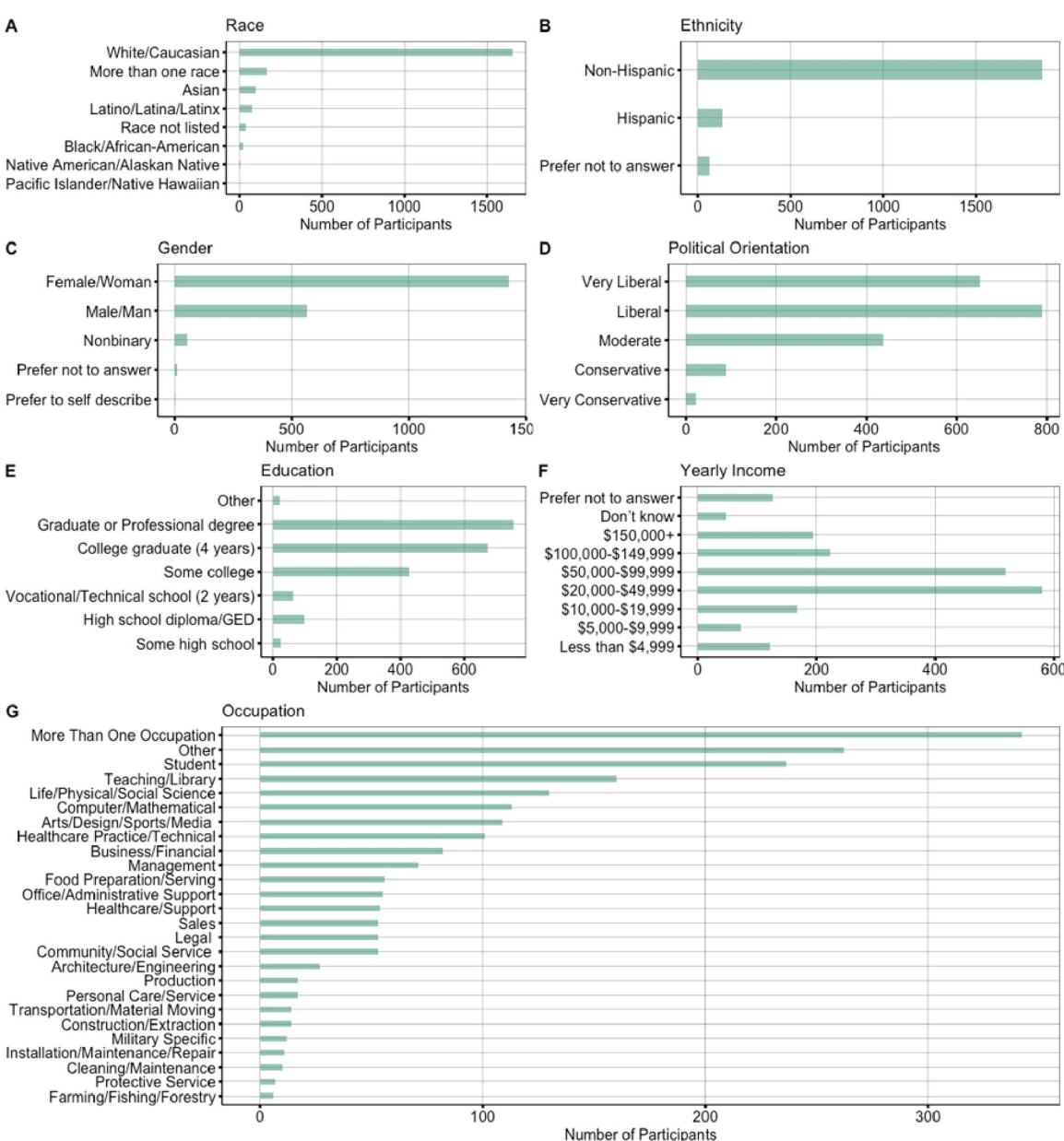

**Fig 2. Description of participant demographics.**

from the general population prior to the COVID-19 pandemic the average GAD-2 score was 0.82 (SD = 1.10) [30].

*Patient Health Questionnaire-2 (PHQ-2).* The PHQ-2 consists of the first two questions of the PHQ-9 and has been shown to be a valid measure of depressive symptoms [26, 31, 32] with a cut-off score $\geq$ 3 for identifying likely Major Depressive Disorder with a sensitivity value of 0.83 and a specificity value of 0.92 [31], although a more recent meta-analysis identified a lower pooled sensitivity value of 0.76 (0.68–0.82) and a lower pooled specificity value of 0.87 (.82–0.90) [32]. In a sample of over 5,000 individuals from the general population prior to the COVID-19 pandemic the average PHQ-2 score was 0.94 (SD = 1.20) [30].

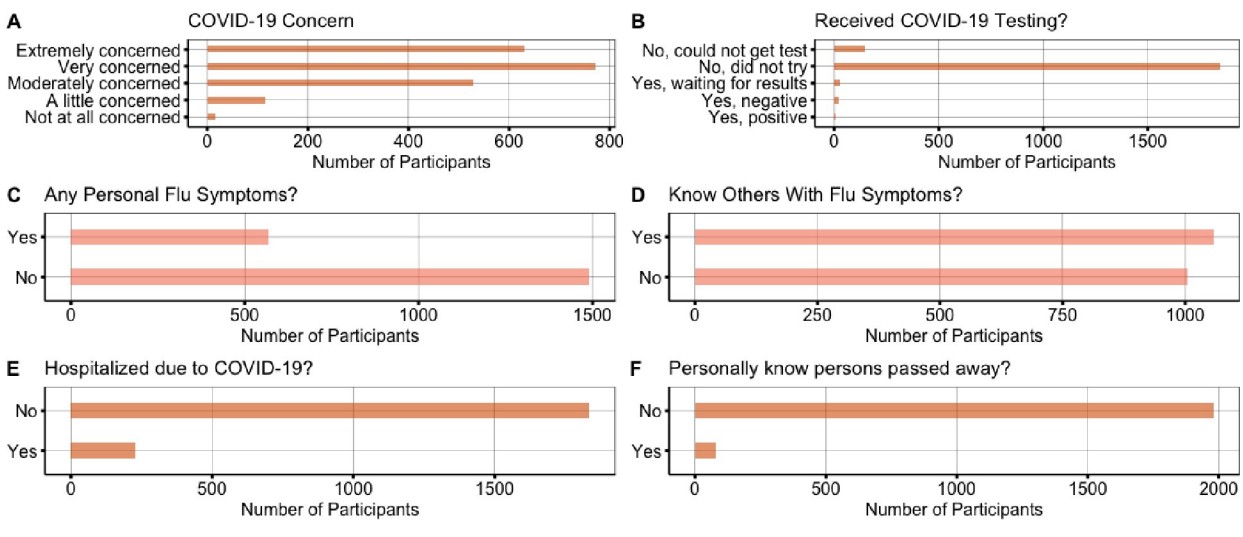

**Fig 3. Description of COVID-19 health variables.**

*COVID-19 concern*. We created a measure of COVID-19 concern that stated, "How concerned do you feel about COVID-19?" with a 5-point Likert scale ranging from "Not at all concerned" to "Extremely concerned."

*COVID-19 measures*. We created a list of COVID-19 measures including items associated with personal flu symptoms, COVID-19 testing, hospitalization and known relationship with someone with COVID-19 symptoms, testing, hospitalization and/or death, and behavioral questions included change in behavior and self-quarantine due to COVID-19. Specifically, we assessed change in behavior by asking, "Have you made any changes to your daily lifestyle due to COVID-19?" with the following responses: a. Yes, I have made changes to my daily schedule to reduce risk or b. No, I have not made changes to my daily schedule to reduce risk. Furthermore, we assessed self-quarantine by asking, "How much are you self-quarantining?" with the following responses: a. None of the time. I am continuing my normal daily schedule, b. Some of the time. I have reduced some of the time that I am in public spaces, social gatherings, and work, c. Most of the time. I only leave for food, doctor appointments, and other essentials, or d. All of the time. I am staying home almost all of the time. For the complete list of questions and item responses, please see OSF (https://osf.io/vtnca/).

*Financial strain*. We created a measure of COVID-associated financial strain included questions associated with lost or change in job, income, and financial comfort. Annual income (prior to COVID-19) and highest education was also obtained.

## Epidemiological data

**Confirmed cases, deaths, recovered.** Epidemiological data on confirmed cases were extracted from the Johns Hopkins University Center for Systems Science and Engineering github (https://github.com/CSSEGISandData/COVID-19). Data on daily confirmed world and country cases were used and merged with participant data on date that participant filled out the questionnaire.

## Statistical analyses

All statistical analyses were conducted with R Studio, version 1.1.463. See S1 File on Open Science Framework (OSF; https://osf.io/vtnca/) for statistical code and packages used for analyses.

Statistical significance was defined using 95% confidence intervals and *p*-values. A series of multilevel model delineating correlations between measures of interest were used to assess 1) the associations between transdiagnostic mental health symptoms and objective epidemiological risk (i.e., confirmed world and country cases) with COVID-19 concern; 2) the associations between financial strain, transdiagnostic mental health symptoms, and COVID-19 concern; and 3) the associations between COVID-19 concern, transdiagnostic mental health symptoms, and objective epidemiological risk (i.e., confirmed world and country cases) with degree of adherence to self-quarantine recommendations. For each model, intercepts were allowed to vary by country, and models controlled for age, gender, and date participants filled out the questionnaire. For each analysis we ran a set of models 1) unadjusted and adjusted transdiagnostic mental health symptoms and 2) unadjusted and adjusted epidemiological variables before 3) a final single and more stringent model was run including transdiagnostic mental health symptoms, epidemiological variables, and covariates in order to identify which specific variable accounted for the most variance in the model. In addition, bar chart figures were created with the ggstatsplot package [33] to visually depict the percent of participants reaching anxiety and depression diagnostic cutoff scores by level of COVID-19 concern. Confirmed world and country cases were log transformed to correct for skew. See OSF for tables of results presented below.

## Results

### Descriptives

Descriptive statistics revealed high levels of COVID-19 concern, such that that 30.51% (n = 630) were extremely concerned, 37.43% (n = 773) were very concerned, 25.62% (n = 529) were moderately concerned, 5.57% (n = 115) were a little concerned, and 0.73% (n = 15) were not at all concerned. Furthermore, 27.61% (n = 569) of participants reported experiencing flu symptoms, while 51.31% (n = 1058) of participants reported knowing someone that was exhibiting flu symptoms. Lastly, 10.94% (n = 226) of the participants were hospitalized due to COVID-19 and 3.83% (n = 79) personally knew of someone that had passed away due to COVID-19 (see Fig 3).

Descriptives also revealed that 98.50% of participants (n = 2034) had made lifestyle changes with varying degrees of self-quarantine, such that 41.50% (n = 857) spending all the time, 51.14% (n = 1056) most of the time, 6.39% (n = 132) some of the time, and 0.77% (n = 16) none of the time in self-quarantine (see Fig 4).

Lastly, 32.88% (n = 679) of participants had lost income and 13.56% (n = 280) had lost their job due to COVID-19. In terms of financial security for each month, 38.50% (n = 795) of participants reported being comfortable with extra, 37.09% (n = 766) reported having enough, but no extra, 19.13% (n = 395) reporting they had to cut back, and 5.13% (n = 106) reporting that they could not make ends meet. In terms of those reporting food security in the last 12 months (e.g., whether they ran out of food and didn't have money to buy more), 91.33% (n = 1886) reported that this was never true, 7.41% (n = 153) reported sometimes true, and 1.11% (n = 23) reported often true (see Fig 5).

### Anxiety and depressive symptoms compared to historical averages

In the current study and as shown in Fig 6, the average GAD-2 score was 3.31 (SD = 1.97) and the average PHQ-2 score was 2.59 (SD = 1.80), which indicates significant elevations in anxiety symptoms, $t(2061) = 57.287$, $p < .001$, 95% CI (3.221, 3.392), and depressive symptoms, $t(2061) = 41.717$, $p < .001$, 95% CI (2.516, 2.671) during the COVID-19 pandemic as compared to past normative data from the general population where the average GAD-2 score was 0.82 (SD = 1.10) and the average PHQ-2 score was 0.94 (SD = 1.20) [30].

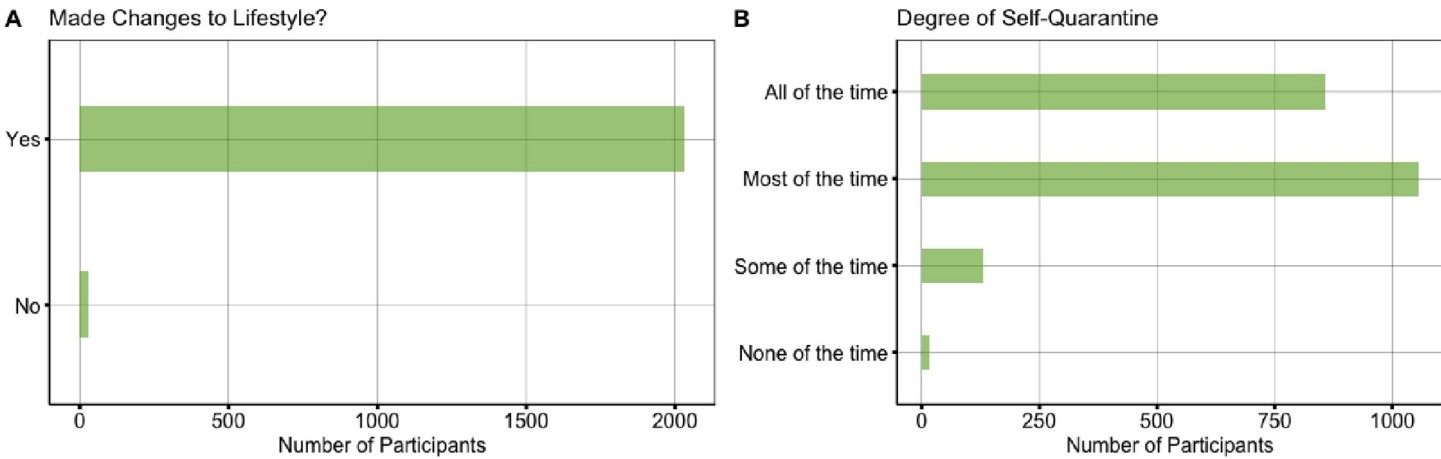

**Fig 4. Behavioral changes due to COVID-19.**

## Mental health and epidemiological correlates of COVID-19 concern

**Mental health correlates of COVID-19 concern.** Results showed that greater anxiety (B = 0.219, SE = 0.009, $p < 0.001$, 95% CI [0.201 – 0.236]) and depressive symptoms (B = 0.151, SE = 0.011, $p < 0.001$, 95% CI [0.133 – 0.175]) were both positively associated with COVID-19 concern. When anxiety and depressive symptoms were entered into the same model, greater anxiety symptoms were significantly positively associated with COVID-19 concern (B = 0.211, SE = 0.012, $p < 0.001$, 95% CI [0.188 – 0.233]), while depressive symptoms were not (B = 0.014, SE = 0.013, $p = 0.283$, 95% CI [-0.011 – 0.039]), indicating that anxiety symptoms are a more strongly related to COVID-19 concern than depressive symptoms. Fig 7 displays percent of participants that met cutoff score by level of COVID-19 concern.

**Epidemiological correlates of COVID-19 concern.** Results indicated that higher confirmed world cases at time survey was filled out (B = 0.665, SE = 0.332, $p = 0.045$, 95% CI [0.014 – 1.316]) and higher confirmed country cases at time survey was filled out (B = 0.053, SE = 0.018, $p = 0.003$, 95% CI [0.018 – 0.088]) were also positively associated

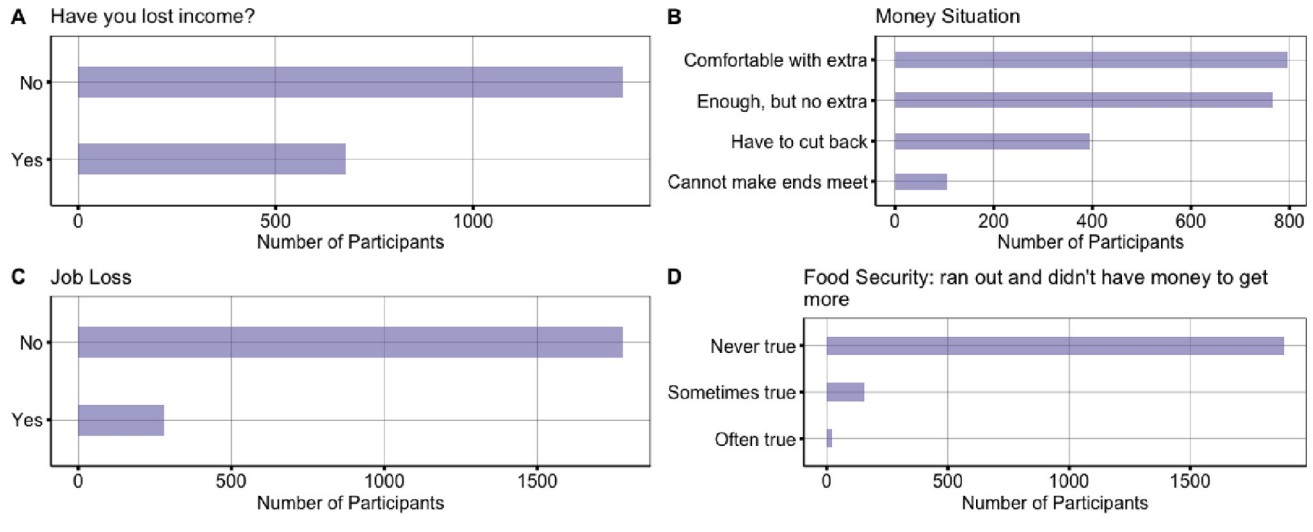

**Fig 5. Financial strain.**

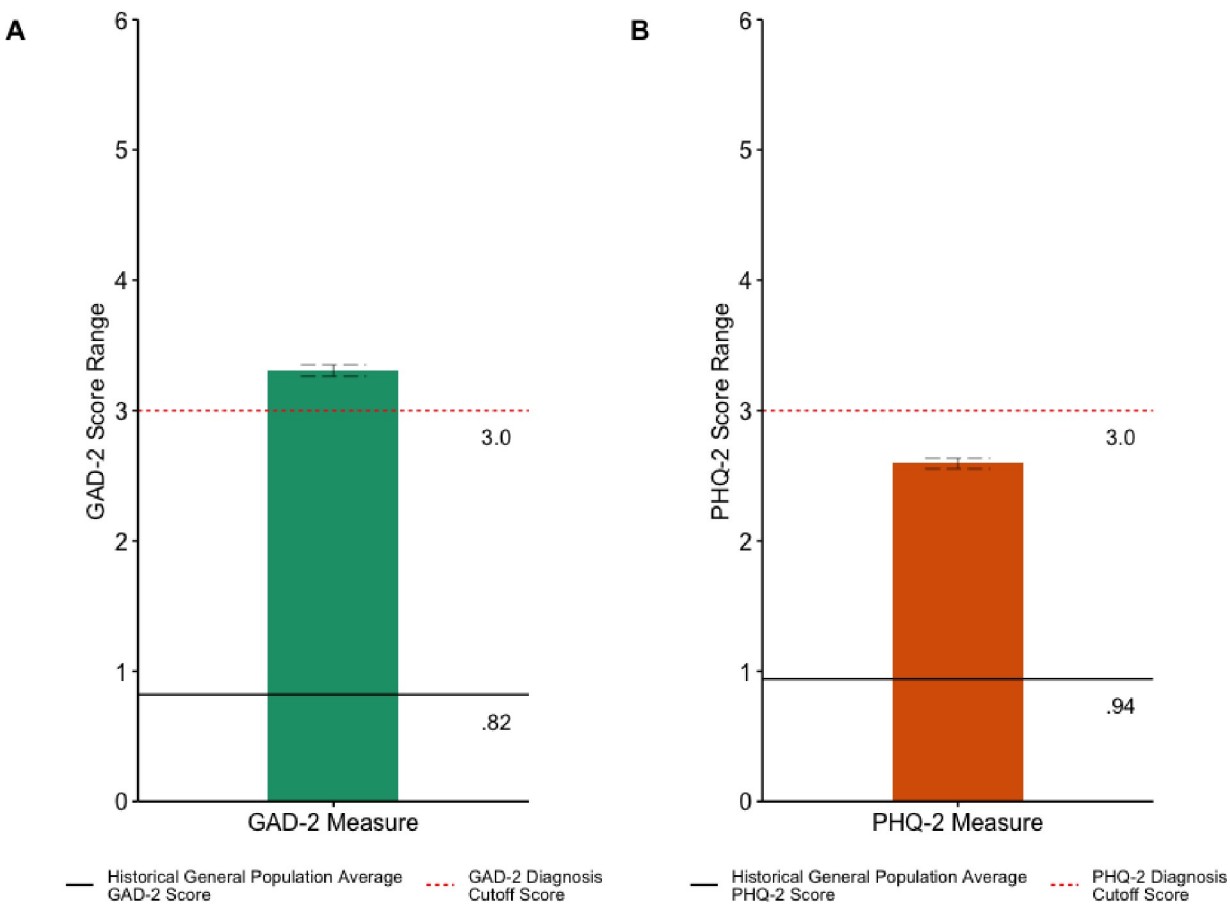

**Fig 6.** Mean a) GAD-2 and b) PHQ-2 Total Scores and Clinical Diagnostic Cutoff Compared to Historical General Population Mean Total Scores. Note: Solid Black Line = GAD-2 and PHQ-2 Historical General Population Mean Total Scores; Dotted Red Line = Threshold for Clinical Diagnosis; Grey Dashed Line error bars = Standard Error; GAD-2 = Generalized Anxiety Disorder-2; PHQ-2 = Patient Health Questionnaire-2; *** = $p < .001$.

with COVID-19 concern. When world and country confirmed cases were entered into the same model, higher country confirmed cases (B = 0.042, SE = 0.019, $p$ = 0.015, 95% CI [0.006 – 0.079]), but not world confirmed cases (B = 0.584, SE = 0.343, $p$ = 0.089, 95% CI [-0.088 – 1.256]) were significantly associated with increased COVID-19 concern, indicating that regional cases are more strongly positively associated with COVID-19 concern when compared to global cases.

**Combined mental health and epidemiological model.** Lastly, when both transdiagnostic mental health symptoms and epidemiological data of confirmed world and country cases were included in the same model, greater anxiety was significantly positively associated with higher COVID-19 concern (B = 0.209, SE = 0.012, $p$ < 0.001, 95% CI [0.186 – 0.232]). In contrast, depressive symptoms (B = 0.014, SE = 0.013, $p$ = 0.259, 95% CI [-0.011 – 0.039]), confirmed world cases (B = 0.289, SE = 0.303, $p$ = 0.340, 95% CI [-0.304 – 0.882]), and confirmed country cases (B = 0.018, SE = 0.017, $p$ = 0.283, 95% CI [-0.015 – 0.050]) was not significantly associated with COVID-19 concern. It is important to note that older age (range 18–77) was also positively associated with increased COVID-19 concern (B = 0.014, SE = 0.002, $p$ < 0.001, 95% CI [0.011 – 0.018]).

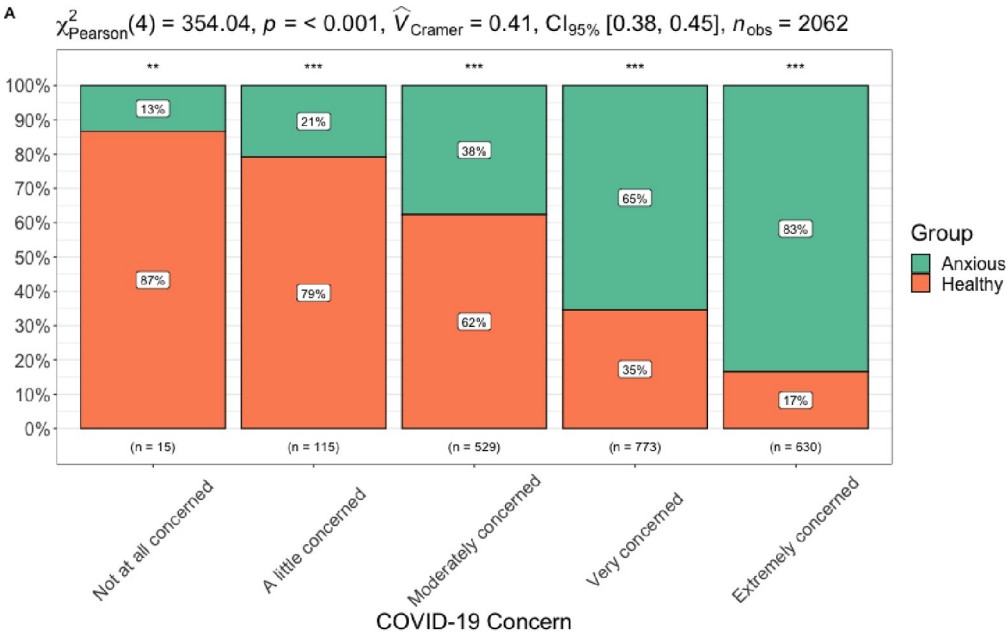

In favor of null: $\log_e(BF_{01}) = -172.44$, $a = 1.00$

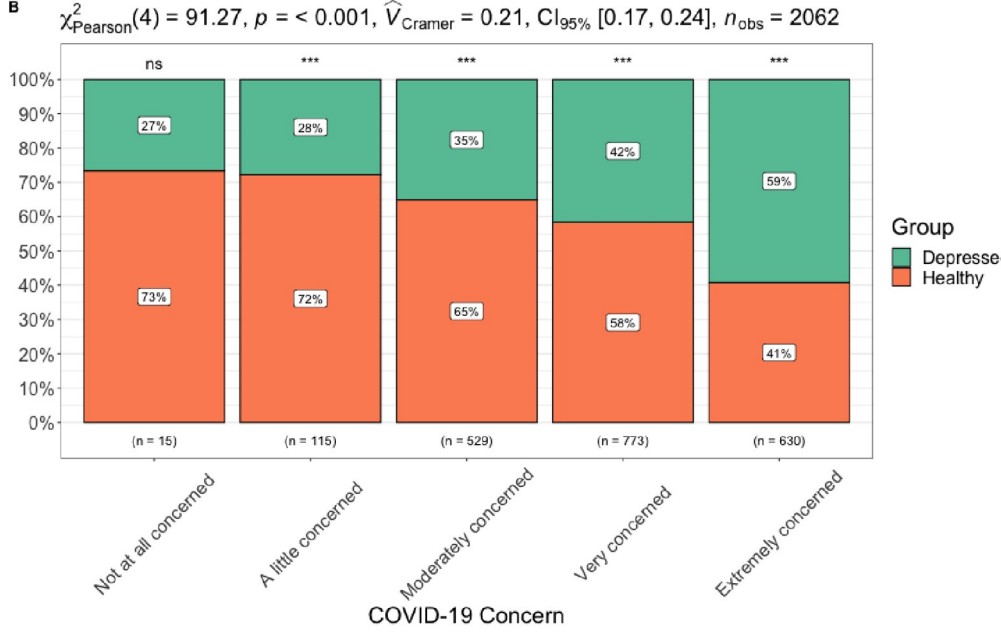

In favor of null: $\log_e(BF_{01}) = -35.29$, $a = 1.00$

**Fig 7.** A. GAD-2 and B. PHQ-2 diagnostic threshold by level of COVID-19 concern.

## Relationship between financial strain, mental health, and COVID-19 concern

Loss of employment was positively associated with greater COVID-19 concern (B = 0.179, SE = 0.042, $p < 0.001$, 95% CI [0.096 – 0.262]), greater depressive symptoms (B = 0.456, SE = 0.084, $p < 0.001$, 95% CI [0.291 – 0.622]) and greater anxiety symptoms (B = 0.346, SE = 0.093, $p < 0.001$, 95% CI [0.165 – 0.528]).

### Correlates of adherence to stay at home orders

**Anxiety and depressive symptom correlates of self-quarantine behavioral adherence.**
Greater anxiety (B = 0.032, SE = 0.007, $p < 0.001$, 95% CI [0.018 – 0.046]) and depressive
(B = 0.040, SE = 0.008, $p < 0.001$, 95% CI [0.025 – 0.056]) symptoms were significantly related
to adherence to more stringent self-quarantine recommendations. When anxiety and depres-
sive symptoms were entered into the same model, greater depressive symptoms (B = 0.029,
SE = 0.010, $p = 0.003$, 95% CI [0.010 – 0.049]) were positively associated with adherence to
more stringent self-quarantine recommendations, but anxiety symptoms were not (B = 0.016,
SE = 0.009, $p = 0.077$, 95% CI [-0.002 – 0.034]).

**COVID-19 concern correlates of self-quarantine behavioral adherence.** Greater
COVID-19 concern was positively associated with more strict adherence to self-quarantine
recommendations (B = 0.136, SE = 0.016, $p < 0.001$, 95% CI [0.106 – 0.167]).

**Epidemiological correlates of self-quarantine behavioral adherence.** Neither confirmed
country (B = 0.005, SE = 0.011, $p = 0.633$, 95% CI [-0.017 – 0.028]) nor world cases (B = 0.341,
SE = 0.236, $p = 0.148$, 95% CI [-0.121 – 0.804]) were associated with more strict behavioral
quarantine recommendations.

**Combined mental health, COVID-19 concern, and epidemiological model.** Lastly,
when transdiagnostic mental health symptoms, COVID-19 concern, and Epidemiological var-
iables of confirmed world and country cases were included in the same model greater depres-
sive symptoms (B = 0.028, SE = 0.010, $p = 0.005$, 95% CI [0.008 – 0.047]) and COVID-19
concern (B = 0.132, SE = 0.018, $p < 0.001$, 95% CI [0.097 – 0.166]) were significantly positively
associated with increased degree of adherence to self-quarantine recommendations. In con-
trast, anxiety symptoms (B = -0.012, SE = 0.010, $p = 0.219$, 95% CI [-0.031 – 0.007]), confirmed
country cases (B = -0.001, SE = 0.011, $p = 0.913$, 95% CI [-0.024 – 0.021]), and confirmed
world cases (B = 0.225, SE = 0.235, $p = 0.338$, 95% CI [-0.236 – 0.686]) were not significantly
positively associated with degree of adherence to self-quarantine recommendations.

## Discussion

The current study recruited 2,065 participants across the United States, Canada, and Europe to
investigate the initial impact the COVID-19 pandemic on the level of psychological concern
about the pandemic, mental health symptoms, financial stability, and degree of adherence to
self-quarantine health behavior.

Data provided compelling evidence of increased anxiety and depression symptoms com-
pared to historical normative data, indicating a clinically significant increase in societal mental
health difficulties. These finding converge with prior research on local epidemic and disease
outbreaks [5, 7, 8, 10] and are consistent with recent poll data from 150,000 Americans [13].

Results also indicated that individual differences in mental health symptoms and epidemio-
logical data signifying objective world/country confirmed cases were significantly positively
associated with increased COVID-19 concern, although when all variables were placed into
the same model, then anxiety symptoms were the strongest correlate of COVID-19 concern.
Although our sample skewed toward higher education and was predominantly White, our
findings through online recruitment were consistent with recent and more representative poll
data of over 16,000 American individuals indicating COVID-19 concern [34]. In the current
study models, age was significantly positively associated with COVID-19 concern, which is
consistent with increased case severity and CFR for older individuals [35].

Loss of employment was positively associated with increased COVID-19 concern and men-
tal health symptoms, the latter of which has been previously documented during global reces-
sions [21]. These results coincide with historical increases in unemployment [17, 18], likely

increasing concern related to COVID-19 as individuals not only worry about immediate health, but also secondary economic implications, yet the data collected were cross-sectional, which precludes any ability to make directional claims with data from the current study.

Last, depressive symptoms and COVID-19 concern were both related to having more stringent self-quarantine behaviors, while this was not true of epidemiological variables representing confirmed world and country cases, which may partially relate to objective risk for participants in specific areas around the world. These finding coincide with past research indicating that emotional risk perceptions are often stronger determinants of behavior change than objective risk [36, 37] as well as research showing that successful public health interventions for 2009 H1N1 Swine Flu were impacted by individual risk perception, indicating that risk perception is a critical driver of protective behavior [38]. Alternatively, it is possible that individuals that were under more stringent self-quarantine, for whatever reason, may have higher depressive symptoms due to the strain of social isolation and the lack of social interactions as depression is associated with social withdrawal. Similarly, loss of employment was associated with depressive symptoms and perhaps the reason these participants were quarantining was because they had nowhere to go. Again, data presented here were correlational and in no way allow us to make directional claims. Future longitudinal studies will be required to parse apart directionality.

These findings may inform public health interventions designed to slow infection. Given that there is currently no vaccine for the virus, the most effective intervention is population wide self-quarantine and "physical distancing". Conformity with these guidelines, however, is effortful, economically damaging, and conflicts with the powerful human motivation for social contact. As such, understanding correlates of adherence to guidelines is essential information for informing more effective public health campaigns. Findings presented here indicate that although public education about objective measures of infection and death rates, especially at the country level, do correlate with psychological concern with COVID-19, the strongest associations of both COVID-19 concern and conformity with self-quarantine measures are measures of individual differences in mental health symptoms. These findings, while correlational that in no way imply causation, may suggest that the propensity for worry and sadness is somewhat adaptive in the current environment during which circumstances are objectively threatening and defensive behavior is in both personal and public interest. However, previous fear-based behavior change campaigns have been controversial [39]. It is likely that some level of fear and sadness is adaptive in objectively threatening circumstances, as long as it's not severe enough to induces behavioral paralysis, and is combined with self-efficacy [40]. Public education may need to focus on low fear and sadness individuals. Furthermore, older age was positively associated with increased COVID-19 concern, so public education may also benefit from focusing increasing self-quarantine among younger aspects of the population as they seem to have lower levels of COVID-19 concern.

## Limitations

While the present study had a number of significant strengths such as rapidly collecting comprehensive psychosocial, health, and economic data on 2,065 participants across the United States, Canada, and Europe during the initial stages of the COVID-19 global pandemic, there are a number of limitations to note. First, online recruitment has been found to have variable demographic and political representation [41]. Our sample was overwhelmingly White with political views that tended to lean moderate to left, limiting the ability to generalize findings to individuals of other races and political orientations. This limitation is an important limitation as preliminary data have shown that African Americans have disproportionately contracted

and passed away due to COVID-19 [42–45] and recent poll data have found that 76% of conservatives believe the media has have exaggerated COVID-19 risks, potentially indicating less COVID-19 concern [46]. Second, the study was limited to the United States, Canada, and Europe and therefore was strongly skewed to White individuals, which may preclude these data from generalizing to other non-Western countries. Third, the study was cross-sectional limiting the ability to address changes in mental health symptoms, COVID-19 concern, and financial stability across time. The current study is continuing data collection each month for 12 months, which will allow for longitudinal assessment of the dynamics of psychological adjustment during global pandemics. Fourth, the current study did not collect other psychosocial variables, such as social support and coping that may moderate effects found in the present study. Again, these important data will be collected at follow-ups with the current sample. Fifth, the current study conceptualized confirmed world and country cases with objective risk, which is a potential oversimplification that has the potential to lead to misinterpretation of findings. For example, the current study did not collect data on household presence of, or caretaker responsibilities for, high risk individuals, which would be a key factor that would greatly increase objective risk. Future studies should collect additional factors that would allow researchers to identify those at objectively higher risk for COVID-19 transmission. Sixth, and related to the prior point, results showed that mental health symptoms explained more variance in COVID-19 concern as compared to variables that we defined as indicating objective risk (e.g., confirmed world and country cases). It is possible that other factors related to being in a high risk group (e.g., preexisting medical complications, which themselves highly covary with mental health symptoms) or even differences in personality (e.g., conscientiousness or neuroticism) may have been an unexplained third variable that may have led to higher COVID-19 concern and stringent self-quarantine behaviors. Future research should collect these variables to provide a more comprehensive evaluation of participant behavior change.

## Conclusion

Our cross-sectional study recruited 2,065 participants across the United States, Canada, and Europe to investigate whether current levels of anxiety and depressive symptoms are elevated compared to historical normative data, determine the strength of psychological and epidemiological associations with COVID-19 concern, identify correlates of financial strain, and identify associations with these variables and engagement in more stringent self-quarantine behaviors. Findings indicated that current anxiety and depressive symptoms are elevated compared to historical norms, these mental health symptoms explain more variance in COVID-19 concern when compared to epidemiological data signifying confirmed world and country cases. In addition, loss of employment was positively associated with greater depressive symptoms and COVID-19 concern, and that COVID-19 concern and depressive symptoms explained the most variance in adhering to more stringent self-quarantine behavior, which have implications for slowing the spread of the COVID-19 pandemic.

## Supporting information

**S1 File.**
(DOCX)

## Acknowledgments

We thank all participants who contributed data to and spread the word about this study.
  Dedication

The manuscript is dedicated to all healthcare workers and scientists on the frontlines working to take care of patients and to identify means to slow the spread of COVID-19.

## Author Contributions

**Conceptualization:** Benjamin W. Nelson, Jessica E. Flannery.

**Data curation:** Benjamin W. Nelson.

**Formal analysis:** Benjamin W. Nelson, Adam Pettitt, Jessica E. Flannery.

**Funding acquisition:** Nicholas B. Allen.

**Methodology:** Benjamin W. Nelson.

**Supervision:** Nicholas B. Allen.

**Visualization:** Benjamin W. Nelson, Adam Pettitt.

**Writing – original draft:** Benjamin W. Nelson, Jessica E. Flannery, Nicholas B. Allen.

**Writing – review & editing:** Benjamin W. Nelson, Adam Pettitt, Jessica E. Flannery, Nicholas B. Allen.

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
