## [Decision Letter · Decision Letter 0]

13 Jul 2020

PONE-D-20-11961

Rapid Assessment of Psychological and Epidemiological Correlates of COVID-19 Concern, Financial Strain, and Health-Related Behavior Change in a Large Online Sample

PLOS ONE

Dear Dr. Nelson,

Thank you for submitting your manuscript to PLOS ONE. After careful consideration, we feel that it has merit but does not fully meet PLOS ONE’s publication criteria as it currently stands. Therefore, we invite you to submit a revised version of the manuscript that addresses the points raised during the review process.

We look forward to receiving your revised manuscript.

Kind regards,

Vincenzo De Luca

Academic Editor

PLOS ONE

Journal Requirements:

2.We note that Figure 1 in your submission contains map/satellite images which may be copyrighted.

We require you to either (a) present written permission from the copyright holder to publish these figure specifically under the CC BY 4.0 license, or (b) remove the figure from your submission:

b. If you are unable to obtain permission from the original copyright holder to publish these figure under the CC BY 4.0 license or if the copyright holder’s requirements are incompatible with the CC BY 4.0 license, please either i) remove the figure or ii) supply a replacement figure that complies with the CC BY 4.0 license. Please check copyright information on all replacement figures and update the figure caption with source information. If applicable, please specify in the figure caption text when a figure is similar but not identical to the original image and is therefore for illustrative purposes only.

3. Please note that according to our submission guidelines (http://journals.plos.org/plosone/s/submission-guidelines), outmoded terms and potentially stigmatizing labels should be changed to more current, acceptable terminology.

For example: “Caucasian” should be changed to “white” or “of [Western] European descent” (as appropriate).

4. We note that you have indicated in your Ethics Statement that "The current study was IRB human subject exempt." However, line 148/149 of the manuscript indicates that "This study was approved by the University of Oregon Institutional Review Board.".

Please correct and clarify.

5.Thank you for stating the following in the Acknowledgments Section of your manuscript:

'This research was supported by funding from Ann Swindells Endowment to the University of Oregon.'

'No authors have any financial disclosures.'

5. Please include captions for your Supporting Information files at the end of your manuscript, and update any in-text citations to match accordingly. Please see our Supporting Information guidelines for more information: http://journals.plos.org/plosone/s/supporting-information

Reviewers' comments:

Reviewer's Responses to Questions

**Comments to the Author**

1. Is the manuscript technically sound, and do the data support the conclusions?

Reviewer #1: Partly

Reviewer #2: Partly

2. Has the statistical analysis been performed appropriately and rigorously? 

Reviewer #1: I Don't Know

Reviewer #2: Yes

3. Have the authors made all data underlying the findings in their manuscript fully available?

Reviewer #1: Yes

Reviewer #2: Yes

4. Is the manuscript presented in an intelligible fashion and written in standard English?

Reviewer #1: Yes

Reviewer #2: Yes

5. Review Comments to the Author

Reviewer #1: This review is of a manuscript under consideration for publication in PLOS-One by Nelson et al., entitled “Rapid assessment of psychological and epidemiological correlates of COVID-19 concern, financial strain, and health-related behavior change in a large online sample.” The paper describes a cross-sectional study conducted by self-report of Americans during late March/early April 2020, during the coronavirus (COVID-19) pandemic. The authors conclude that mental health symptoms are more highly associated with COVID-19-related concern and quarantine behaviors and make leaps of interpretation regarding possible impacts on behavioral changes. The paper should benefit from rapid publication, due to the sensitivity of the timing of information related to the pandemic; however, the reviewer does find major concerns that must be addressed prior to publication. With that in mind, the reviewer wishes to stress that there are no “fatal flaw” concerns here; the listed problems should be able to be addressed without extensive additional work, and the length of the comments are primarily intended to make clear the reviewer’s thoughts.

Major comments:

1. The paper equates confirmed world and or country cases with objective risk, an oversimplification that leads to an overreach of conclusions and a possible misinterpretation of findings. The survey does not allow the authors, as far as the reviewer can tell, to measure household presence of, or caretaker responsibilities for, high risk individuals, a key factor that greatly modifies objective risk in this scenario and which affects a large number of individuals. Caretakers for, or individuals of, high risk must be more vigilant than the average person under normal circumstances and thus, were almost certainly more attuned to the danger of COVID-19 early in the pandemic. Additionally, it was known fairly early that age of over 60 or 65 increased risk, greatly widening the number of people falling into the high risk category and the number of people directly affected by the health of high-risk individuals. Indeed, the authors report that age is positively correlated with COVID-19 concern, making this scenario possible. The question is, are individuals affected by higher risk driving the findings relating to stringent quarantine behavior and or concern about the virus? The authors conclude that mental health symptoms (anxiety and depression) explain more variance in COVID-19 concern than objective risk, but using world/country case numbers to indicate objective risk is not an equivalent representation for a large number of people. The authors say their findings coincide with studies suggesting that emotional risk perceptions are stronger determinants of behavior change than objective risk. However, if their findings are driven by high risk-affected individuals, who are ostensibly at greater risk for infection, complications and death, this interpretation isn’t appropriate. If by late March/early April, the U.S. had still had a negligible number of cases, perhaps equating country cases to objective risk would be fine here, but that was not the case. The reviewer suggests a reconsideration of interpretations with this in mind. At the very least, there should be space dedicated to this possibility in the discussion and the authors’ longer-term follow up should explore this avenue. The authors should be careful to also change abstract and conclusion wording accordingly to best reflect their final interpretation.

2. The paper implies that correlation, or association, implies causation and without much discussion of other possibilities, they imply a direction of causation. Specifically, they say “These findings suggest that the propensity for worry and sadness is somewhat adaptive in the current environment…”, and suggest that fear and sadness are causing protective behaviors. Related to the reviewers concerns in the separate point above, it could be that individuals at high risk or with direct ties to or caretaking roles for high risk individuals (their children and/or parents) are experiencing mental health symptoms and are taking the virus more seriously due to their more prominent, objective risk. Likewise, the authors seem to favor that perhaps depressive symptoms are improving quarantine behavior, but another interpretation is that individuals under more stringent quarantine, for whatever reason, have increased symptoms of depression due to strain of isolation, including a lack of social interactions. The reviewer suggests that the authors think more carefully about the implications of their work; they may wish to keep some of their points about public education, but need to round those arguments out carefully and consider other implications their work may have and how that information could impact interventions for the public.

3. There are many spelling, grammatical, and other errors that should be corrected to meet the quality standards of the journal and to best convey the information.

a. Some examples:

i. “inflection” is found at least twice, instead of “infection”

ii. Case fatality rate (CFR) should be defined properly on Line 63. Later “CFR” is used alone.

iii. Line 85: “recent preprints have found” – should be something like “studies in recent preprints show”

iv. Line 90 “casing” – should be “causing”; also the “and” before “casing” should be removed, as it is not a complete sentence as is

v. Line 100 – perhaps missing word “been”

vi. Line 110 – change “do” to “due”

vii. Lines 110-114 – awkward sentence. Suggest, “Furthermore, the impact of social isolation and loneliness on physical health is on par with that of…”

viii. Line 115-120

1. use of “due to” twice in same sentence

2. “and” is placed before #3 and #4

3. Also, this is a gigantic sentence that needs to be broken into two separate sentences

ix. Line 122-124 – awkward sentence

x. Line 124-125 – should be “This cross-sectional study…”

xi. Line 135 – if number will be used to start a sentence, should be spelled out

xii. Line 141 – should be “IP address”

xiii. Line 141 – when referring to people, it is “who” or “whom” instead of “that”

xiv. Not certain about format used for some citations (e.g., Reddit, ProPublica, CDC, Johns Hopkins).

xv. Line 172-174 – awkward sentence uses “and has been shown to be” twice

xvi. Some sentences start with “This” and do not reference a noun. Example Line 199-200.

xvii. “epidemiological of risk” is used a few times – not clear what is meant, maybe a typo? Examples Line 217, Line 213.

xviii. Line 259-263 – run-on sentence (“which indicates…which indicates…” within same sentence)

xix. Line 303 – “associate” should be “associated”

xx. Line 413 – “…with of COVID-19 concern…” – need to remove “of”

xxi. Line 414 – need to add the word “with” – sentence not complete

4. At times, the manuscript leaves out important details, and in general, would benefit from re-wording to convey information more accurately. Listed are instances of concern as examples, and it would likely be beneficial if the authors would use the same eye to re-examine the manuscript a bit more broadly.

a. Introduction: “Overall, some epidemiological models have predicted that there will be 1.1 – 1.2 million deaths in the United States and 250,000 deaths in Great Britain (Ferguson et al., 2020).” This sentence comes at the end of a paragraph that does a nice job of covering ranges of estimates concerning COVID-19’s R0 and fatality rate. Then suddenly, the authors use an unmitigated estimate of deaths, citing Ferguson et al., which is meant to model mortality in the “… absence of any control measures or spontaneous changes in individual behaviour…”. There are other relevant considerations of the model, including that it does not account “for the potential negative effects of health systems being overwhelmed on mortality.” The authors need to include more of this information to give the reader an accurate sense of its relevance, particularly in light of the moderate way they have approached the other estimates.

b. Line 71: Is use of “Spanish flu” conventional in the field? Perhaps “1918 influenza pandemic” or “1918 H1N1 pandemic” or other wording is more accurate and informative.

c. Line 85: “…increased Google mental health searches…” Consider rewording to say “internet mental health-related keyword searches using the Google search engine,” or similar.

d. The authors use “in modern times” in multiple places. Use of this phrase is probably fine, but more meaning would be gleaned if they gave a little more information, such as “since a [particular year or event].” The example in Line 99, “…leading to the largest decrease in world trade and GDP in modern times.” There are major events last century (WWII, Great Depression, etc.) that are familiar to most readers—is there a semi-recent event that can be accurately pointed to instead as a comparison?

e. “Associated” is used throughout the manuscript, but “positively” or “negatively” correlated or similar would be more informative.

i. Example Line 105

f. Lines 136-141 – sentence is too long and implies that limiting participation to certain countries led to removing participants based on age, etc. Suggest starting a new sentence after “Europe (n = 245)” on Line 139.

g. Related to above, Lines 139-141 concerning how participants were excluded is confusing. The authors simultaneously describe inclusion criteria (“18 or older”) and the number of individuals removed for not meeting those criteria (“n=12”), so that it seems like 12 people were kept in because they were 18 or older. Also consider the following sentence where inclusion criteria are mentioned again. This information can be more streamlined.

h. Mean age and range should include the unit (years)

i. Line 200 – “year income” – probably meant “annual income”

j. Line 354-356 – confusing sentence – suggest, “Our sample demographic was skewed toward higher education and was predominantly Caucasian, consistent with online recruitment studies assessing larger…”

k. The very next sentence starts with “in these models” and it is unclear to the reader whether the authors mean the larger online recruitment studies that they just referred to or their own models in the current paper.

l. Lines 354-356 seem out of place. If Lines 356-358 do refer to the larger studies they should be moved with Lines 354-356; see last comment)

5. General corrections need to be made

i. Some figures have legend-style descriptions (e.g., Figs. 6, 7), while others have very minimal information (only a title?) (e.g., Figs. 2, 3, 4). Some figure legends use “a)” and others “A”, etc.

ii. Figure 7 has “covid_concern” in panel A, which seems like it should be updated to “Covid-19 Concern” similar to panel B.

iii. more stats explanation is needed

1. the authors say they used models and give a link to the statistical code, which will be good for future use, but the reader has little idea how to evaluate these statistical analyses. Throughout the paper, the authors say that a particular variable is “associated” with an outcome, but is their finding similar to a correlation? If so, can they say positively or negatively associated with…? For example, Lines 304-305, “…older age (range 18-77) was associated with COVID-19 concern (B = 0.014, SE = …). More description about the statistical models used would be helpful.

2. what is the “B” value given throughout?

These comments are also uploaded as a separate file for the authors.

Reviewer #2: The study is well designed with a good rationale and good choice of brief symptom measures. The manuscript is exceptionally well written, clear and easy to follow. For the most part, I have only minor suggestions.

Main Issue: The main conceptual issue that I would like to see addressed relates to the interpretation of the findings regarding greater depression symptoms being associated with adherence to more stringent self-quarantine recommendations, despite anxiety symptoms being the strongest correlate of COVID-19 concern. This begs the question as to whether depression is really driving adherence to COVID-19 restrictions per se, or if it’s a spurious correlation related to depressive behavior, not COVID-19 related behavior. Under normal circumstances depression is be associated with social withdrawal. It appears that people in the sample who lost their jobs were also more depressed. Perhaps the reason they are “quarantining” is because they lost their job and have no where to go? To help me understand this better, I wondered specifically how adherence to COVID-19 restrictions was assessed. For example, is this staying home/self-isolating only, or does the “change in behavior” also include COVID-19 specific behaviors such as wearing a mask in public, physical distancing, cleaning hands frequently? If it is only staying home, this may be better explained by depression and job loss and not really specific to COVID-19 which is implied. I tried to find supplemental information specifying the questions used to measure “change in behavior” and “self-quarantine due to COVID-19” however these measures did not appear to be included in the supplemental material attached to the submission, nor were the questions easily located at the provided OSF website. Please clarify in the manuscript how "change in behavior" and "self-quarantine due to COVID-19" was measured.

Related to the above:

Line 307 – loss of employment was associated with both greater COVID-19 concern, greater depressive symptoms and greater anxiety symptoms. If you entered loss of employment into the same model with depressive symptoms and anxiety symptoms, do you get the same results as when you have depression and anxiety symptoms alone? (Line 317-320)

Minor Points

Future directions – consider including measures of personality, in particular the Big 5 Personality traits (NEO-PI). One wonders if personality traits such as conscientiousness or neuroticism might also predict adherence to COVID-19 restrictions?

Line 286 – Figure 7. I don’t think you can use dx = diagnosis when you only had 2 items from the PHQ. It’s an indicator of depressive symptoms that has been shown to be sensitive and specific, but on it’s own would not be sufficient to make a diagnosis.

Line 386 – do you think public education should be specifically geared towards younger population since your results also found that older age was associated with increased COVID-19 concern? (Line 305) ie, younger age associated with lower concern?

6. PLOS authors have the option to publish the peer review history of their article (what does this mean?). If published, this will include your full peer review and any attached files.

Reviewer #1: **Yes: **Laura Smith

Reviewer #2: No

---

## [Author Response · Author response to Decision Letter 0]

4 Sep 2020

Reviewer #1: This review is of a manuscript under consideration for publication in PLOS-One by Nelson et al., entitled “Rapid assessment of psychological and epidemiological correlates of COVID-19 concern, financial strain, and health-related behavior change in a large online sample.” The paper describes a cross-sectional study conducted by self-report of Americans during late March/early April 2020, during the coronavirus (COVID-19) pandemic. The authors conclude that mental health symptoms are more highly associated with COVID-19-related concern and quarantine behaviors and make leaps of interpretation regarding possible impacts on behavioral changes. The paper should benefit from rapid publication, due to the sensitivity of the timing of information related to the pandemic; however, the reviewer does find major concerns that must be addressed prior to publication. With that in mind, the reviewer wishes to stress that there are no “fatal flaw” concerns here; the listed problems should be able to be addressed without extensive additional work, and the length of the comments are primarily intended to make clear the reviewer’s thoughts.

Major comments:

1. The paper equates confirmed world and or country cases with objective risk, an oversimplification that leads to an overreach of conclusions and a possible misinterpretation of findings. The survey does not allow the authors, as far as the reviewer can tell, to measure household presence of, or caretaker responsibilities for, high risk individuals, a key factor that greatly modifies objective risk in this scenario and which affects a large number of individuals. Caretakers for, or individuals of, high risk must be more vigilant than the average person under normal circumstances and thus, were almost certainly more attuned to the danger of COVID-19 early in the pandemic. Additionally, it was known fairly early that age of over 60 or 65 increased risk, greatly widening the number of people falling into the high risk category and the number of people directly affected by the health of high-risk individuals. Indeed, the authors report that age is positively correlated with COVID-19 concern, making this scenario possible. The question is, are individuals affected by higher risk driving the findings relating to stringent quarantine behavior and or concern about the virus? The authors conclude that mental health symptoms (anxiety and depression) explain more variance in COVID-19 concern than objective risk, but using world/country case numbers to indicate objective risk is not an equivalent representation for a large number of people. The authors say their findings coincide with studies suggesting that emotional risk perceptions are stronger determinants of behavior change than objective risk. However, if their findings are driven by high risk-affected individuals, who are ostensibly at greater risk for infection, complications and death, this interpretation isn’t appropriate. If by late March/early April, the U.S. had still had a negligible number of cases, perhaps equating country cases to objective risk would be fine here, but that was not the case. The reviewer suggests a reconsideration of interpretations with this in mind. At the very least, there should be space dedicated to this possibility in the discussion and the authors’ longer-term follow up should explore this avenue. The authors should be careful to also change abstract and conclusion wording accordingly to best reflect their final interpretation.

 Thank you for this important point. As requested, we have now added the conceptualization of confirmed world and country cases with objective risk to the limitations section. Specifically, we have stated,

“Fifth, the current study conceptualized confirmed world and country cases with objective risk, which may be an oversimplification that has the potential to lead to misinterpretation of findings. For example, the current study did not collect data on household presence of, or caretaker responsibilities for, high risk individuals, which would be a key factor that would significantly increase objective risk. Future studies should collect additional data such as these that would allow researchers to identify those at objectively higher risk for COVID-19 transmission.”

In terms of the possibility that age is driving the association with stringent quarantine behaviors, participant age was included in the model and was not a significant covariate, indicating that this aspect of being high risk may not have translated to more stringent quarantine behaviors. None-the-less, we have added the following to the limitations section:

“Sixth, and related to the prior point, results showed that mental health symptoms explained more variance in COVID-19 concern than did variables that we defined as indicating objective risk (e.g., confirmed world and country cases). It is possible that other factors related to being in a high risk group (e.g., preexisting medical complications, which themselves highly covary with mental health symptoms) may have been an unexplained third variable that may have led to higher COVID-19 concern and stringent self-quarantine behaviors. Future research should collect these variables to provide a more comprehensive evaluation of participant behavior change.”

Lastly, we have changed our abstract and conclusion wording to describe that we compared mental health symptoms to epidemiological data signifying confirmed world and country cases, rather than use language about objective risk.

2. The paper implies that correlation, or association, implies causation and without much discussion of other possibilities, they imply a direction of causation. Specifically, they say “These findings suggest that the propensity for worry and sadness is somewhat adaptive in the current environment…”, and suggest that fear and sadness are causing protective behaviors. Related to the reviewers concerns in the separate point above, it could be that individuals at high risk or with direct ties to or caretaking roles for high risk individuals (their children and/or parents) are experiencing mental health symptoms and are taking the virus more seriously due to their more prominent, objective risk. Likewise, the authors seem to favor that perhaps depressive symptoms are improving quarantine behavior, but another interpretation is that individuals under more stringent quarantine, for whatever reason, have increased symptoms of depression due to strain of isolation, including a lack of social interactions. The reviewer suggests that the authors think more carefully about the implications of their work; they may wish to keep some of their points about public education, but need to round those arguments out carefully and consider other implications their work may have and how that information could impact interventions for the public.

 Thank you for this important point. We have tempered our language in the discussion to highlight the correlational nature of the findings and explicitly state that our findings in no way can imply causation. Specifically, in regards to the quote that you cite, we state:

“These findings, while correlational that in no way imply causation, may suggest if confirmed in a longitudinal study that the propensity for worry and sadness is somewhat adaptive in the current environment, where circumstances are objectively threatening and defensive behavior is in both personal and public interest.”

 As proposed by the reviewers, we have also added in other possibilities for results (e.g., association between depressive symptoms and quarantine behaviors) indicating that directionality cannot be determined in this study due to the cross-sectional nature of the study design. Specifically, we state,

“Alternatively, it is possible that individuals that were under more stringent self-quarantine, for whatever reason, may have higher depressive symptoms due to the strain of social isolation and the lack of social interactions. Again, data presented here were correlational and in no way allow us to make directional claims. Future longitudinal studies will be required to parse apart directionality.”

Although we appreciate the reviewer’s suggestion that we expand on the implications of this study, we are also mindful of the additional material we have already added in response to other reviewer comments, and as such we have been conservative about responding to this issue. However, as suggested by Reviewer 2, we have added some material about the implications of the findings for targeting public education towards specific age groups. 

3. There are many spelling, grammatical, and other errors that should be corrected to meet the quality standards of the journal and to best convey the information.

a. Some examples:

i. “inflection” is found at least twice, instead of “infection”

This has been edited.

ii. Case fatality rate (CFR) should be defined properly on Line 63. Later “CFR” is used alone.

This has been edited.

iii. Line 85: “recent preprints have found” – should be something like “studies in recent preprints show”

This has been edited

iv. Line 90 “casing” – should be “causing”; also the “and” before “casing” should be removed, as it is not a complete sentence as is

This has been edited.

v. Line 100 – perhaps missing word “been”

This has been edited

vi. Line 110 – change “do” to “due”

This has been edited

vii. Lines 110-114 – awkward sentence. Suggest, “Furthermore, the impact of social isolation and loneliness on physical health is on par with that of…”

This has been edited

viii. Line 115-120

1. use of “due to” twice in same sentence

This has been edited

2. “and” is placed before #3 and #4

This has been edited

3. Also, this is a gigantic sentence that needs to be broken into two separate sentences

Sentence reduced in size

ix. Line 122-124 – awkward sentence

This has been edited

x. Line 124-125 – should be “This cross-sectional study…”

This has been edited

xi. Line 135 – if number will be used to start a sentence, should be spelled out

This has been edited

xii. Line 141 – should be “IP address”

This has been edited

xiii. Line 141 – when referring to people, it is “who” or “whom” instead of “that”

This has been edited

xiv. Not certain about format used for some citations (e.g., Reddit, ProPublica, CDC, Johns Hopkins).

Thank you, we have checked formatting.

xv. Line 172-174 – awkward sentence uses “and has been shown to be” twice

This has been edited

xvi. Some sentences start with “This” and do not reference a noun. Example Line 199-200.

This has been edited.

xvii. “epidemiological of risk” is used a few times – not clear what is meant, maybe a typo? Examples Line 217, Line 213.

This has been edited

xviii. Line 259-263 – run-on sentence (“which indicates…which indicates…” within same sentence)

This has been edited

xix. Line 303 – “associate” should be “associated”

This has been edited

xx. Line 413 – “…with of COVID-19 concern…” – need to remove “of”

This has been edited

xxi. Line 414 – need to add the word “with” – sentence not complete

This has been edited

4. At times, the manuscript leaves out important details, and in general, would benefit from re-wording to convey information more accurately. Listed are instances of concern as examples, and it would likely be beneficial if the authors would use the same eye to re-examine the manuscript a bit more broadly.

a. Introduction: “Overall, some epidemiological models have predicted that there will be 1.1 – 1.2 million deaths in the United States and 250,000 deaths in Great Britain (Ferguson et al., 2020).” This sentence comes at the end of a paragraph that does a nice job of covering ranges of estimates concerning COVID-19’s R0 and fatality rate. Then suddenly, the authors use an unmitigated estimate of deaths, citing Ferguson et al., which is meant to model mortality in the “… absence of any control measures or spontaneous changes in individual behaviour…”. There are other relevant considerations of the model, including that it does not account “for the potential negative effects of health systems being overwhelmed on mortality.” The authors need to include more of this information to give the reader an accurate sense of its relevance, particularly in light of the moderate way they have approached the other estimates.

 Thank you for raising this point. As suggested, we have now added that this is one of the initial epidemiological models that is based on an “unmitigated epidemic” (e.g., absence of individual behavior change and systemic control measures) that doesn’t account for the potential of overwhelmed healthcare systems.

b. Line 71: Is use of “Spanish flu” conventional in the field? Perhaps “1918 influenza pandemic” or “1918 H1N1 pandemic” or other wording is more accurate and informative.

 Thank you for this suggestion. We have reworded this according to your suggestion.

c. Line 85: “…increased Google mental health searches…” Consider rewording to say “internet mental health-related keyword searches using the Google search engine,” or similar.

 We have rephrased the sentence as suggested. 

d. The authors use “in modern times” in multiple places. Use of this phrase is probably fine, but more meaning would be gleaned if they gave a little more information, such as “since a [particular year or event].” The example in Line 99, “…leading to the largest decrease in world trade and GDP in modern times.” There are major events last century (WWII, Great Depression, etc.) that are familiar to most readers—is there a semi-recent event that can be accurately pointed to instead as a comparison?

 We have made appropriate edits to provide names of major events, rather than referring to “modern times.”

e. “Associated” is used throughout the manuscript, but “positively” or “negatively” correlated or similar would be more informative.

 We have revised our wording to include the directionality of associations throughout the manuscript.

i. Example Line 105

 We have revised our wording to include the directionality of the association in this example.

f. Lines 136-141 – sentence is too long and implies that limiting participation to certain countries led to removing participants based on age, etc. Suggest starting a new sentence after “Europe (n = 245)” on Line 139.

 We have revised this sentence by starting a new sentence as suggested.

g. Related to above, Lines 139-141 concerning how participants were excluded is confusing. The authors simultaneously describe inclusion criteria (“18 or older”) and the number of individuals removed for not meeting those criteria (“n=12”), so that it seems like 12 people were kept in because they were 18 or older. Also consider the following sentence where inclusion criteria are mentioned again. This information can be more streamlined.

 Thank you for these points. We have revised this sentence to indicate that we removed 12 subjects who were younger than 18 years of age, which was required for our inclusion criteria. We have also clarified the other exclusion criteria.

h. Mean age and range should include the unit (years) 

 We have now specified that age is in years.

i. Line 200 – “year income” – probably meant “annual income”

 This has been revised as suggested.

j. Line 354-356 – confusing sentence – suggest, “Our sample demographic was skewed toward higher education and was predominantly Caucasian, consistent with online recruitment studies assessing larger…”

 This has been edited as suggested. 

k. The very next sentence starts with “in these models” and it is unclear to the reader whether the authors mean the larger online recruitment studies that they just referred to or their own models in the current paper.

 We have added language to clarify this point by stating,

“In the current study models, age was significantly positively associated with COVID-19 concern, which is consistent with increased case severity and CFR for older individuals (CDC, 2020b).”

l. Lines 354-356 seem out of place. If Lines 356-358 do refer to the larger studies they should be moved with Lines 354-356; see last comment)

 Thank you for this suggestion. We have reworded this sentence.

5. General corrections need to be made

i. Some figures have legend-style descriptions (e.g., Figs. 6, 7), while others have very minimal information (only a title?) (e.g., Figs. 2, 3, 4). Some figure legends use “a)” and others “A”, etc.

 The figures with extra information were automatically generated by a separate package to that used to generate the other descriptive figures. The package is meant to visualize different statistical models and therefore provides information that is not readily available in (nor necessarily relevant to) the package used to make the other figures. If the reviewer provides the additional statistics or information that they consider to be important to display on the other figures, then the authors are happy to provide that information.

ii. Figure 7 has “covid_concern” in panel A, which seems like it should be updated to “Covid-19 Concern” similar to panel B.

 We have made the change to Figure 7.

iii. more stats explanation is needed

 The authors are more than happy to provide extra explanations for the statistics used, but would need more specific guidance on what the reviewer would like to be included. For example, some of the figures are merely presentations of descriptive data, rather than statistical analyses per se, so we are unsure what else to include at this time. If the reviewer would be willing to identify which concepts need clarification, we are more than willing to add these to the manuscript. 

1. the authors say they used models and give a link to the statistical code, which will be good for future use, but the reader has little idea how to evaluate these statistical analyses. Throughout the paper, the authors say that a particular variable is “associated” with an outcome, but is their finding similar to a correlation? If so, can they say positively or negatively associated with…? For example, Lines 304-305, “…older age (range 18-77) was associated with COVID-19 concern (B = 0.014, SE = …). More description about the statistical models used would be helpful.

 We have now provided a description that we used multilevel models to delineate correlations between measures of interest. Specifically, we state, 

“A series of multilevel model delineating correlations between measures of interest were used to assess 1) the associations between transdiagnostic mental health symptoms and objective epidemiological risk (i.e., confirmed world and country cases) with COVID-19 concern; 2) the associations between financial strain, transdiagnostic mental health symptoms, and COVID-19 concern; and 3) the associations between COVID-19 concern, transdiagnostic mental health symptoms, and objective epidemiological risk (i.e., confirmed world and country cases) with degree of adherence to self-quarantine recommendations.”

In the results, we have also added directionality every time we mention any associations to highlight that the statistical models were looking at correlations.

2. what is the “B” value given throughout?

 The B values are the beta values or the degree of change in the outcome variable for every 1 unit of change in the predictor variable.

 

Reviewer #2: The study is well designed with a good rationale and good choice of brief symptom measures. The manuscript is exceptionally well written, clear and easy to follow. For the most part, I have only minor suggestions.

Main Issue: The main conceptual issue that I would like to see addressed relates to the interpretation of the findings regarding greater depression symptoms being associated with adherence to more stringent self-quarantine recommendations, despite anxiety symptoms being the strongest correlate of COVID-19 concern. This begs the question as to whether depression is really driving adherence to COVID-19 restrictions per se, or if it’s a spurious correlation related to depressive behavior, not COVID-19 related behavior. Under normal circumstances depression is be associated with social withdrawal. It appears that people in the sample who lost their jobs were also more depressed. Perhaps the reason they are “quarantining” is because they lost their job and have no where to go? To help me understand this better, I wondered specifically how adherence to COVID-19 restrictions was assessed. For example, is this staying home/self-isolating only, or does the “change in behavior” also include COVID-19 specific behaviors such as wearing a mask in public, physical distancing, cleaning hands frequently? If it is only staying home, this may be better explained by depression and job loss and not really specific to COVID-19 which is implied. I tried to find supplemental information specifying the questions used to measure “change in behavior” and “self-quarantine due to COVID-19” however these measures did not appear to be included in the supplemental material attached to the submission, nor were the questions easily located at the provided OSF website. Please clarify in the manuscript how "change in behavior" and "self-quarantine due to COVID-19" was measured.

We strongly agree with the reviewer’s suggestion and have added an alternative explanation for the finding of an association between depressive symptoms and self-quarantine. Specifically, we state,

“Alternatively, it is possible that individuals that were under more stringent self-quarantine, for whatever reason, may have higher depressive symptoms due to the strain of social isolation and the lack of social interactions as depression is associated with social withdrawal. Similarly, loss of employment was associated with depressive symptoms, and perhaps the reason these participants were quarantining was because they had nowhere to go. Again, data presented here were correlational and in no way allow us to make directional claims. Future longitudinal studies will be required to parse apart directionality.”

 We also addressed the issue of self-quarantining by asking, “How much are you self-quarantining?” with the following responses. 

 a. None of the time. I am continuing my normal daily schedule.

b. Some of the time. I have reduced some of the time that I am in public spaces, social gatherings, and work.

c. Most of the time. I only leave for food, doctor appointments, and other essentials.

d. All of the time. I am staying home almost all of the time.

In addition, we assessed behavior change by asking, “Have you made any changes to your daily lifestyle due to COVID-19?” with the following responses.

 a. Yes, I have made changes to my daily schedule to reduce risk.

 b. No, I have not made changes to my daily schedule to reduce risk.

Therefore, we did assess physical distancing, but did not assess wearing a mask in public or cleaning hands frequently.

Lastly, we have now included the questions related to behavior change and self-quarantine to the methods section and uploaded a list of all study questions to OSF.

Related to the above:

Line 307 – loss of employment was associated with both greater COVID-19 concern, greater depressive symptoms and greater anxiety symptoms. If you entered loss of employment into the same model with depressive symptoms and anxiety symptoms, do you get the same results as when you have depression and anxiety symptoms alone? (Line 317-320)

We ran the requested analysis (i.e., with loss of employment in the same model with depressive symptoms) and anxiety symptoms had the same result as when we had depression and anxiety symptoms alone. Also, we found that loss of employment was significantly related to COVID-19 concern (p = 0.021).

Minor Points

Future directions – consider including measures of personality, in particular the Big 5 Personality traits (NEO-PI). One wonders if personality traits such as conscientiousness or neuroticism might also predict adherence to COVID-19 restrictions?

 We have now included this in our limitations section. Specifically, we state,

“Sixth, and related to the prior point, results showed that mental health symptoms explained more variance in COVID-19 concern as compared to variables that we defined as indicating objective risk (e.g., confirmed world and country cases). It is possible that other factors related to being in a high risk group (e.g., preexisting medical complications, which themselves highly covary with mental health symptoms) or even differences in personality (e.g., conscientiousness or neuroticism) may have been an unexplained third variable that may have led to higher COVID-19 concern and stringent self-quarantine behaviors. Future research should collect these variables to provide a more comprehensive evaluation of participant behavior change.”

Line 286 – Figure 7. I don’t think you can use dx = diagnosis when you only had 2 items from the PHQ. It’s an indicator of depressive symptoms that has been shown to be sensitive and specific, but on its own would not be sufficient to make a diagnosis.

 We have changed the figure legend to reflect an anxious and depressed group rather than a diagnosis per se. 

Line 386 – do you think public education should be specifically geared towards younger population since your results also found that older age was associated with increased COVID-19 concern? (Line 305) ie, younger age associated with lower concern?

 Thank you for this great point! We agree and have added the following language to the discussion section,

“Furthermore, older age was positively associated with increased COVID-19 concern, so public education may also benefit from focusing increasing self-quarantine among younger aspects of the population as they seem to have lower levels of COVID-19 concern.”

---

## [Decision Letter · Decision Letter 1]

26 Oct 2020

Rapid Assessment of Psychological and Epidemiological Correlates of COVID-19 Concern, Financial Strain, and Health-Related Behavior Change in a Large Online Sample

PONE-D-20-11961R1

Dear Dr. Nelson,

We’re pleased to inform you that your manuscript has been judged scientifically suitable for publication and will be formally accepted for publication once it meets all outstanding technical requirements.

Kind regards,

Vincenzo De Luca

Academic Editor

PLOS ONE

Additional Editor Comments (optional):

Reviewers' comments:

Reviewer's Responses to Questions

**Comments to the Author**

1. If the authors have adequately addressed your comments raised in a previous round of review and you feel that this manuscript is now acceptable for publication, you may indicate that here to bypass the “Comments to the Author” section, enter your conflict of interest statement in the “Confidential to Editor” section, and submit your "Accept" recommendation.

Reviewer #1: All comments have been addressed

Reviewer #2: All comments have been addressed

2. Is the manuscript technically sound, and do the data support the conclusions?

Reviewer #1: Yes

Reviewer #2: Yes

3. Has the statistical analysis been performed appropriately and rigorously? 

Reviewer #1: Yes

Reviewer #2: Yes

4. Have the authors made all data underlying the findings in their manuscript fully available?

Reviewer #1: Yes

Reviewer #2: Yes

5. Is the manuscript presented in an intelligible fashion and written in standard English?

Reviewer #1: Yes

Reviewer #2: Yes

6. Review Comments to the Author

Reviewer #1: The reviewer appreciates the authors efforts to correct and improve the submitted manuscript and their important contribution to the literature. Concerns were sufficiently mitigated or resolved via author comments/clarifications and through additions and changes to the manuscript. In particular, improvements are noted in changes to the graphical representations and mitigation of wording regarding interpretations. The reviewer recommends to accept the manuscript as is. Below some additional minor comments may be considered by the authors.

•Fig 6. - consider adding “historical” to the population average scores on the graphs (as it is in the legend)

•Fig 7. – the reviewer doesn’t see “dx” in the figure, as is mentioned in the legend

•Line 401 – typo makes meaning difficult to discern; thing the word “and” may need to be removed

•Line 404-411 – The reviewer suggests mild refinement and tightening of the writing in this section. The sentiments conveyed here should be maintained, but the wording is somewhat confusing and awkward, and generally does not align with the writing capability and tone used in the rest of the manuscript.

Reviewer #2: I am satisfied that my comments based on the first draft of this manuscript have been adequately addressed in the revision of this paper. I could question whether age was included as a predictor in all regressions, as is implied in the discussion but is not clear from the description of the results. It would appear that age was used in a separate regression and not included in the omnibus regression. However, the authors do acknowledge that age is a significant predictor of COVID-19 concern, and aiming interventions at younger populations who may not take precautions as seriously which was my feedback on the original paper. Given the time sensitive nature of this paper, I would not require an additional revision to clarify exactly which regressions age was included in; I would be satisfied to accept the revision as is.

7. PLOS authors have the option to publish the peer review history of their article (what does this mean?). If published, this will include your full peer review and any attached files.

Reviewer #1: **Yes: **Laura N. Smith

Reviewer #2: No

---

## [Editor Report · Acceptance letter]

29 Oct 2020

PONE-D-20-11961R1 

Rapid Assessment of Psychological and Epidemiological Correlates of COVID-19 Concern, Financial Strain, and Health-Related Behavior Change in a Large Online Sample 

Dear Dr. Nelson:

I'm pleased to inform you that your manuscript has been deemed suitable for publication in PLOS ONE. Congratulations! Your manuscript is now with our production department. 

Kind regards, 

on behalf of

Dr. Vincenzo De Luca 

Academic Editor

PLOS ONE